# Pharmacological and pharmacokinetic profile of the novel ocular hypotensive prodrug CKLP1 in Dutch-belted pigmented rabbits

Uttio Roy Chowdhury[1], Rachel A. Kudgus[2], Tommy A. Rinkoski[1], Bradley H. Holman[1], Cindy K. Bahler[1], Cheryl R. Hann[1], Joel M. Reid[2], Peter I. Dosa[3], Michael P. Fautsch[1]*

**1** Department of Ophthalmology, Mayo Clinic, Rochester, MN, United States of America, **2** Department of Oncology Research, Mayo Clinic, Rochester, MN, United States of America, **3** Department of Medicinal Chemistry, Institute for Therapeutics Discovery and Development, University of Minnesota, Minneapolis, MN, United States of America

* Fautsch@mayo.edu

**Data Availability Statement:** All relevant data are within the paper and its Supporting Information files.

## Abstract

Elevated intraocular pressure is the only treatable risk factor for glaucoma, an eye disease that is the leading cause of irreversible blindness worldwide. We have identified cromakalim prodrug 1 (CKLP1), a novel water-soluble ATP-sensitive potassium channel opener, as a new ocular hypotensive agent. To evaluate the pharmacokinetic and safety profile of CKLP1 and its parent compound levcromakalim, Dutch-belted pigmented rabbits were treated intravenously (0.25 mg/kg) or topically (10 mM; 4.1 mg/ml) with CKLP1. Body fluids (blood, aqueous and vitreous humor) were collected at multiple time points and evaluated for the presence of CKLP1 and levcromakalim using a liquid chromatography-mass spectrometry/mass spectrometry (LC-MS/MS) based assay. Histology of tissues isolated from Dutch-belted pigmented rabbits treated once daily for 90 days was evaluated in a masked manner by a certified veterinary pathologist. The estimated plasma parameters following intravenous administration of 0.25 mg/kg of CKLP1 showed CKLP1 had a terminal half-life of 61.8 ± 55.2 min, $T_{max}$ of 19.8 ± 23.0 min and $C_{max}$ of 1968.5 ± 831.0 ng/ml. Levcromakalim had a plasma terminal half-life of 85.0 ± 37.0 min, $T_{max}$ of 61.0 ± 32.0 min and $C_{max}$ of 10.6 ± 1.2 ng/ml. Topical CKLP1 treatment in the eye showed low levels (<0.3 ng/mL) of levcromakalim in aqueous and vitreous humor, and trace amounts of CKLP1 and levcromakalim in the plasma. No observable histological changes were noted in selected tissues that were examined following topical application of CKLP1 for 90 consecutive days. These results suggest that CKPL1 is converted to levcromakalim in the eye and likely to some extent in the systemic circulation.

## Introduction

Glaucoma is a progressive neurodegenerative disorder of the eye and the leading cause of irreversible blindness worldwide. Elevated intraocular pressure (IOP) is the predominant and the only treatable risk factor for the disease. As a result, all current treatment options for glaucoma

**Funding:** NIH grants EY21727 (MPF), EY26490 (MPF); Minnesota Partnership for Biotechnology and Medical Genomics #12.06 (MPF, PID); Minnesota Partnership for Biotechnology and Medical Genomics TPDF #15.01 (MPF, PID); National Center for Advancing Translational Sciences of the National Institutes of Health Award Number UL1TR000114 (PID); Mayo Foundation (MPF).

**Competing interests:** The authors have declared that no competing interests exist

are aimed at lowering IOP [1–4]. Unfortunately, existing medical therapies have mild to severe side effects [2, 5–7] and first line treatment options with prostaglandin analogues are ineffective in up to 25% of patients with primary open-angle glaucoma [8]. Therefore, identification of novel ocular hypotensive agents that have minimal side effect profiles will greatly benefit patients and augment the current therapeutic management strategies for the disease.

Our laboratory has identified a new class of ocular hypotensive agents called ATP-sensitive potassium ($K_{ATP}$) channel openers [9]. Several commercially available $K_{ATP}$ channel openers were shown to lower IOP in ex vivo and in vivo experimental model systems [9–11]. Since none of these openers are water-soluble and hence not suitable for human application, we developed an aqueous soluble prodrug based on the structure of levcromakalim [12]. This prodrug, referred to as cromakalim prodrug 1 [CKLP1, chemically referred to as sodium [(3S,4R)-6-cyano-2,2-dimethyl-4-(2-oxopyrrolidin-1-yl)-chroman-3-yl phosphate] is converted to its active metabolite levcromakalim through in vivo phosphatase cleavage [12]. In animal models, topical treatment of CKLP1 to the eye results in similar IOP reduction as found with levcromakalim [12, 13]. Additionally, CKLP1 treatment targets the distal portion of the trabecular outflow pathway leading to reduction of episcleral venous pressure [13]. Owing to its unique site of action, CKLP1 was found to exhibit significant additive effects when used in conjunction with existing glaucoma medications like latanoprost, timolol and Rho kinase inhibitors in normotensive rabbits [13].

Based on the preclinical data [7, 12–14], CKLP1 is a promising candidate for clinical investigation in human patients. However, the pharmacologic properties and systemic disposition of CKLP1 and its conversion to levcromakalim are currently unknown. Using an liquid chromatography-mass spectrophotometry/mass spectrophotometry (LC-MS/MS)-based assay to detect CKLP1 and levcromakalim simultaneously, we report the pharmacokinetic parameters of CKLP1 following intravenous and topical eye administration of the drug in Dutch-belted pigmented rabbits. The safety profile and systemic distribution of CKLP1 were also evaluated in Dutch-belted pigmented rabbits following once daily instillation of eye drops for 90 days.

## Methods

### Reagents

CKLP1 and levcromakalim were synthesized in our laboratory as previously described (originally referred to as [3S,4R]-2) [12]. For pharmacokinetic studies, flavopiridol was obtained from the Pharmaceutical Resources Branch of the National Cancer Institute, HPLC grade acetonitrile was from Fisher Scientific (Fairlawn, NJ), analytical grade ammonium formate was from MilliporeSigma (St. Louis, MO), and Captiva 0.2 μm polypropylene 96 well filter plates were from Agilent (Santa Clara, CA). Heparin coated blood collection tubes (5 ml) were purchased from Covidien (Mansfield, MA).

### Determination of CKLP1 and levcromakalim concentrations by LC-MS/MS

LC-MS/MS analysis of all fluids were performed using a Waters (Milford, MA) Acquity UPLC BEH C18 column (1.7 μm, 2.1 x 50 mm) with an Agilent EC-C18 pre-column (2.7 μm, 2.1 x 5 mm) and a gradient elution program containing ultra-pure water and acetonitrile with 2.5 mM ammonium formate, at a flow rate of 0.2 ml/min. Flavopiridol was used as an internal standard. The column and auto-sampler temperatures were maintained at 40°C and 4°C, respectively. Source parameters were capillary voltage, 2.6 kV; source temperature, 150°C; desolvation temperature, 500°C; cone gas flow, 150 L/h; and desolvation gas flow, 800 L/h. The cone voltages and collision energies for CKLP1, levcromakalim and flavopiridol were determined by MassLynx-Intellistart software, v4.1, and were 32, 34 and 72 V (cone) and 20, 14 and

30 eV (collision), respectively. The retention times were 4.5 minutes for CKLP1, 5.9 minutes for levcromakalim and 5.7 minutes for flavopiridol.

To validate accurate measurement of CKLP1 and levcromakalim, known concentrations of CKLP1 (5, 10, 25, 50, 100, 250, 500 ng/ml) and levcromakalim (1, 5, 10, 25, 50, 100, 250 ng/ml) were added to human plasma (obtained from the Mayo Clinic Division of Transfusion Medicine under an Institutional Review Board approved protocol) to make calibration curves. Quality control was determined for each run by using known quantities of CKLP1 and levcromakalim to verify that the instrument recorded values consistent with the established standard curve. The effect of carryover was evaluated by injecting blank samples before and after running the standard samples for calibration curves. Unknown samples with signal to noise ratios >10 were recorded as acceptable concentrations. Samples with signal to noise ratios <10 were reported as not detectable (ND). Controls consisted of a double blank (without analytes and internal standard) and a single blank (only internal standard). The linearity of the calibration curves was determined by plotting the ratios of analytes to internal standard peak areas. The curve was fitted by least squares regression with a $1/x$ weighting for both CKLP1 and levcromakalim. Data was collected from 3–6.4 minutes for CKLP1 and 4–8 minutes for both levcromakalim and flavopiridol. Waters MassLynx Intellistart software v4.1 was used for analysis.

## Animal Care, treatment and IOP measurement

All experiments with animals were pre-approved by the Institutional Animal Care and Use Committee of the Mayo Clinic, Rochester, MN and adhered to the recommendations in the Guide for the Care and use of Laboratory Animals of the National Institutes of Health, and the tenets of the ARVO Statement for the Use of Animals in Vision Research. Female Dutch-belted pigmented rabbits (age 7 months) were obtained from Covance Research Products, Inc. (Denver, PA). Rabbits were housed in pairs and kept in a climate controlled room with 12 h light and dark cycles. Animals had unrestrained access to water and standard rabbit food pellets at all times. Upon receipt, animals were acclimated for at least 5 days before initiation of the experiment. IOP was measured with a handheld rebound tonometer (Icare Tonovet, Colonial Medical Supply, Franconia, NH, USA) in conscious and minimally restrained, non-anesthetized rabbits as previously described [13]. Briefly, daily IOP was calculated from the average of 3 independent IOP measurements at 3 different time points each day. These time points corresponded to 1 h, 4 h and 23 h post treatment on any given day. Rabbits that needed to be handled for IOP were subjected to sham IOP measurements for two days prior to initiation of experiment. Rabbits (n = 3) were injected intravenously through the right ear vein with CKLP1 as a bolus dose equal to 0.25 mg/kg body weight. Additional rabbits were treated topically through instillation of 50 μl eye drops of a 10 mM (4.1 mg/ml) solution of CKLP1 in sterile PBS. Topical CKLP1 application was continued once daily (9:30 am every day) for either 1 (n = 3), 4 (n = 3), 8 (n = 3) or 90 days (n = 6). In the 1, 4 and 8 day groups, drug was added to both eyes, whereas in the 90 day group, CKLP1 was added to one eye while the contralateral eye received vehicle (PBS, Corning, Manassas, VA).

## Collection of tissues and fluids for pharmacokinetic and pharmacodynamic studies

For rabbits injected with CKLP1 intravenously, central ear vein blood was collected at 5 min, 15 min, 30 min, 60 min, 2h, 4h, 8h and 24h following dosing (day 1). For the first 4 time points (5, 15, 30 and 60 min), animals were anesthetized with 75 mg/kg ketamine, 5mg/kg xylazine and 1 mg/kg acepromazine injected intramuscularly. Following the 60 min time point, rabbits were given 120 ml of plasmalyte subcutaneously to rehydrate. For the 2, 4, 8, and 24 h time

points, animals were lightly sedated each time with 35 mg/kg ketamine and 5 mg/kg xylazine intramuscularly.

For rabbits receiving daily CKLP1 treatment administered topically to the eye for up to 8 days, central ear vein blood was collected at 5 min, 15 min, 30 min, 60 min, 2 h, 4 h, 8 h and 24 h following topical application of CKLP1 on days 1, 4 and 8. Prior to eye enucleation on days 1, 4, and 8 following collection of the 24 h blood sample, aqueous humor was aspirated from the anterior chamber by inserting a tuberculin syringe anterior to the corneal-scleral junction of euthanized animals. Enucleated eyes were bisected at the equator and vitreous humor was collected and centrifuged for 5 minutes at 5000 rpm. The supernatant was used for all subsequent analysis. Both halves of the eye were immediately placed in 10% neutral buffered formalin until they were processed for sectioning.

In rabbits treated for 90 days, blood, aqueous humor, vitreous humor and selected tissues were collected from rabbits 24 hr post-treatment (n = 3) and 72 hr post-treatment (n = 3). Plasma was separated from blood samples by centrifugation at 3000 rpm for 5 minutes. Brain, heart, skeletal muscle, kidney, liver and lung tissues were isolated from each animal, weighed and frozen for use in CKLP1 and levcromakalim distribution studies. All body fluids and tissues were stored at -80˚C until LC-MS/MS analysis.

### Statistical analysis

IOP values are represented as mean ± standard deviation. IOP differences between treated and control eyes were compared using two-tailed paired Student's $t$-test. Pharmacokinetic parameters ($C_{max}$, $T_{max}$, AUC and concentration of CKLP1 and levcromakalim) across various days following treatment were compared with one-way ANOVA. Criterion for statistically significant difference of means was set at $p < 0.05$. Statistical analyses were performed using Microsoft Excel and its data analysis add-on feature.

## Results

### Detection of CKLP1 and levcromakalim by LC-MS/MS

The concentrations of CKLP1 and levcromakalim in body fluids and tissues were measured using liquid chromatography-mass spectrometry with multiple reaction monitoring. Protonated molecular ion ([M+H]+) intensities for CKLP1, levcromakalim and the internal standard were greatest under positive electrospray ionization. Mass spectra for CKLP1, levcromakalim and flavopiridol showed prominent parent ion [M+H]+ peaks at m/z 367, 287 and 402, respectively, and prominent product ion peaks at m/z 86 for CKLP1 and levcromakalim and at m/z 341 for flavopiridol. Based on these data, quantification was done by monitoring the m/z 366.97>85.95 transition for CKLP1, the m/z 287.01>85.95 transition for levcromakalim and the m/z 401.91>340.89 transition for the internal standard flavopiridol, using the positive electrospray ionization mode. Gradient elution with a mobile phase containing 2.5 mM ammonium formate provided sharp peaks that were devoid of any baseline interferences or ionization suppression for all analytes (Fig 1A). Standard curves of peak area ratios of analyte/internal standard against respective concentrations for calibration standards were linear over the ranges for CKLP1 and levcromakalim, respectively, and yielded $r^2$ values >0.99 (Fig 1B).

### Pharmacokinetic parameters of CKLP1 and levcromakalim in plasma following single intravenous administration

The pharmacokinetics of CKLP1 and levcromakalim were characterized in female Dutch-belted pigmented rabbits following intravenous injection of 0.25 mg/kg CKLP1. A graph of CKLP1

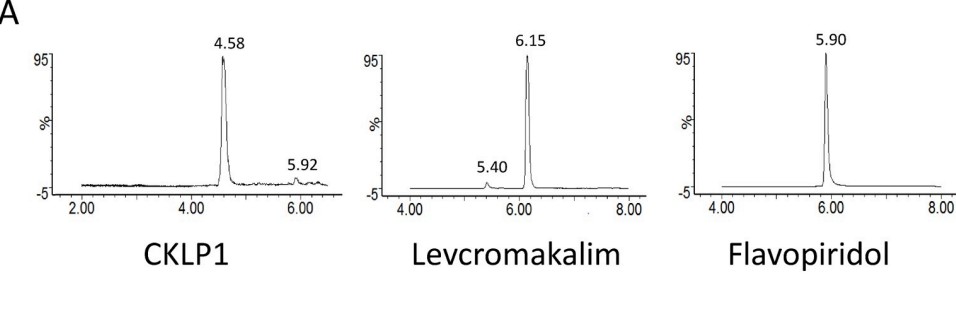

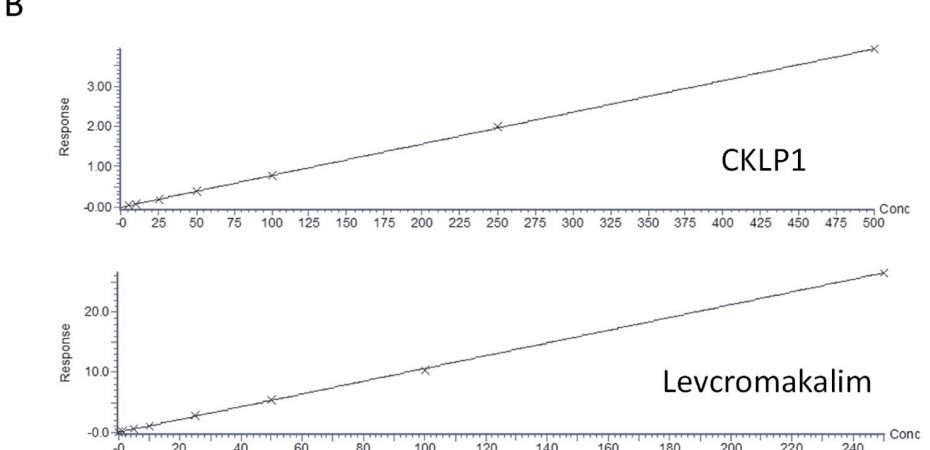

**Fig 1. LC-MS/MS based detection of CKLP1 and levcromakalim.** A. Representative multiple reaction monitoring chromatograms for CKLP1, levcromakalim and the internal standard flavopiridol. B. Standard curves for CKLP1 and levcromakalim were calculated using human plasma containing known amounts of CKLP1 and levcromakalim. Both graphs produced $r^2 > 0.99$.

plasma concentration versus time showed a $C_{max}$ of 1968.5 ± 831.0 ng/ml (n = 6; Table 1) was achieved following the intravenous bolus dose and drug was eliminated from plasma with a half-life of 61.8 ± 55.2 min (Fig 2), $T_{max}$ of 19.8 ± 23.0 min and $C_{max}$ of 1968.5 ± 831.0 ng/ml (n = 6; Table 1). Since CKLP1 is a direct phosphate-linked prodrug based on the structure of levcromakalim [12], we assessed levcromakalim concentrations in plasma. Levcromakalim appearance in plasma was rapid, with a $T_{max}$ of 61.0 ± 32.0 min and $C_{max}$ of 10.6 ± 1.2 ng/ml. The elimination half-life was 85.0 ± 37.0 min (n = 6, Table 1). AUC for CKLP1 was calculated to be 2164.0 ± 482.5 ng/ml*hr and levcromakalim was 120.4 ± 93.7 ng/ml*hr (Table 1).

## Pharmacokinetic analysis of tissue and body fluids following topical application of CKLP1 to the eye

CKLP1 was administered topically to both eyes (50 μl bolus of a 10 mM solution) of Dutch-belted pigmented rabbits for either 1 (n = 3), 4 (n = 3) or 8 days (n = 3). Following treatments,

**Table 1. Pharmacokinetic parameters of CKLP1 and levcromakalim in rabbit plasma following intravenous administration of CKLP1.**

|  | Half-life (min) | $T_{max}$ (min) | $C_{max}$ (ng/ml) | $T_{last}$ (h) | $C_{last}$ (ng/ml) | $AUC_{last}$ (ng/ml*hr) |
|---|---|---|---|---|---|---|
| CKLP1 | 61.8 ± 55.2 | 19.8 ± 23.0 | 1968.5 ± 831.0 | 5.6 ± 1.8 | 34.9 ± 42.4 | 2164.0 ± 482.5 |
| Levcromakalim | 85.0 ± 37.0 | 61.0 ± 32.0 | 10.6 ± 1.2 | 16.0 ± 8.9 | 4.4 ± 4.3 | 120.4 ± 93.7 |

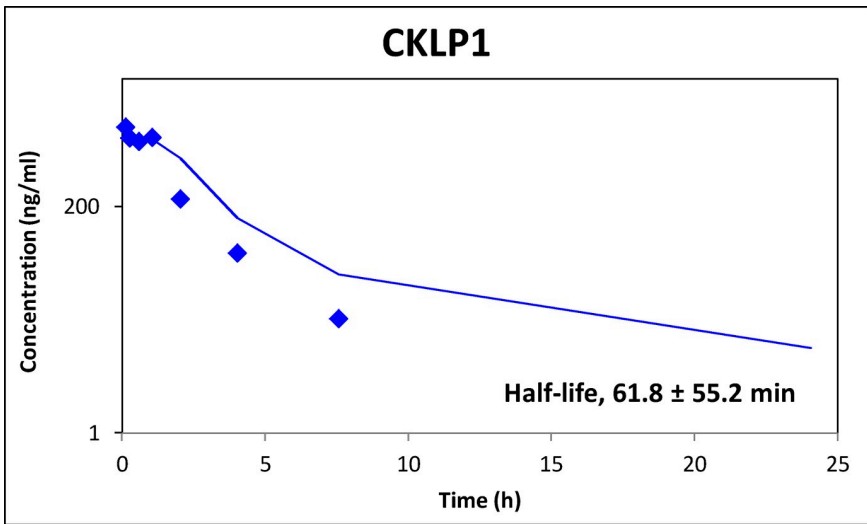

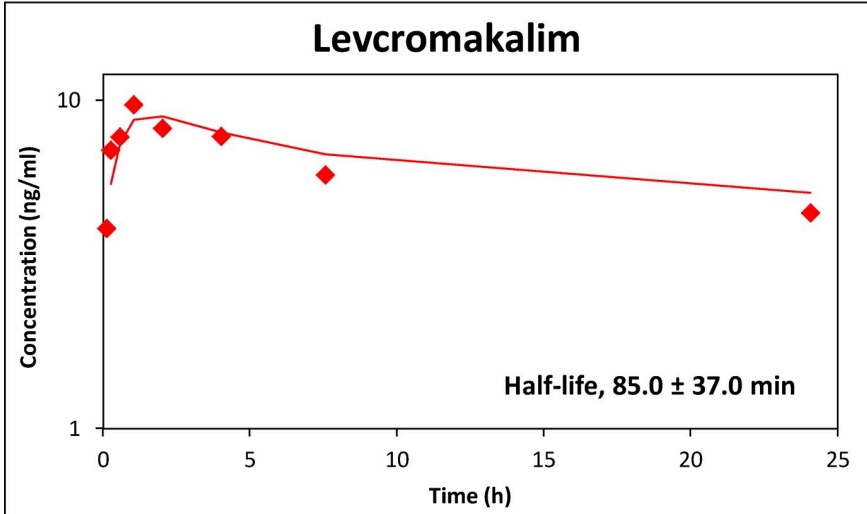

**Fig 2. Plasma concentration of CKLP1 and levcromakalim following intravenous injection of CKLP1.**
Concentrations of CKLP1 were approximately 20X greater than levcromakalim at each time point. Increased levels of levcromakalim were identified in early time points consistent with CKLP1 conversion to levcromakalim in plasma. Both CKLP1 and levcromakalim show characteristic absorption and elimination profiles. The calculated half-life for CKLP1 was $61.8 \pm 55.2$ min and $85.0 \pm 37.0$ min for levcromakalim. Data represented in log scale with a moving average trend line.

blood, aqueous humor and vitreous humor were collected from three animals each on days 1, 4 and 8. Pharmacokinetic analysis on these 3 days showed characteristic distribution and absorption curves of CKLP1 and levcromakalim (Fig 3). In general, the peak concentrations were higher on days 4 and 8 for both CKLP1 and levcromakalim compared to day 1 (Fig 3). CKLP1 and levcromakalim were detected in plasma on each day. Since CKLP1 was administered topically, concentrations in plasma of both CKLP1 and levcromakalim were much lower than what was seen with IV administration. Following administration on day 1, $C_{max}$ was $20.0 \pm 6.3$ ng/ml for CKLP1 and $0.4 \pm 0.1$ ng/ml for levcromakalim. Distribution to plasma was slow with $T_{max}$ values of $360.0 \pm 207.9$ min and $400.0 \pm 138.6$ min, respectively. $C_{max}$ values were found to be higher on days 4 ($30.6 \pm 7.3$ ng/ml for CKLP1 and $0.9 \pm 0.7$ ng/ml for levcromakalim) and 8 ($36.3 \pm 13.3$ ng/ml for CKLP1 and $0.6 \pm 0.3$ for levcromakalim) compared

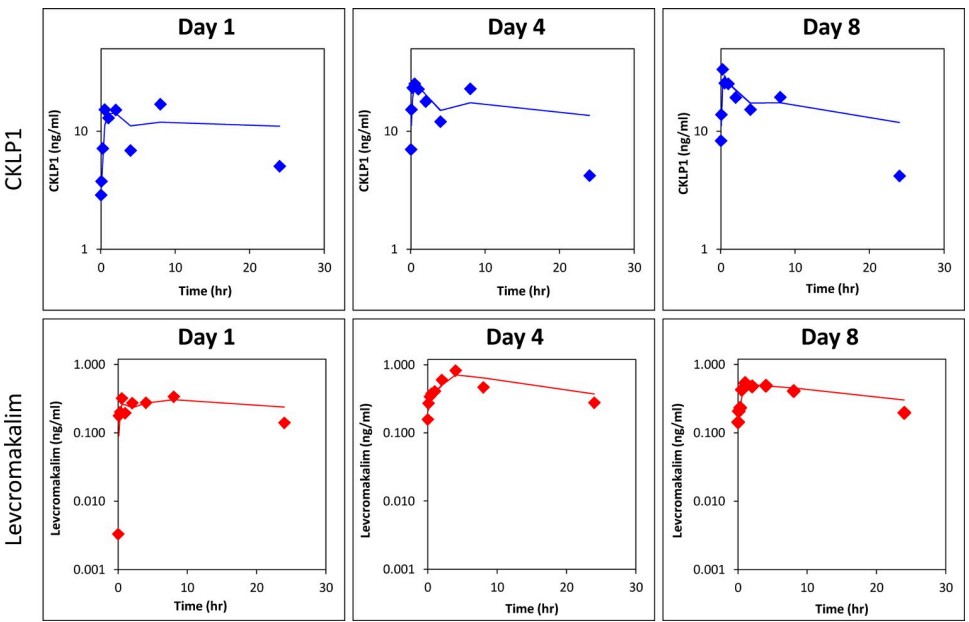

**Fig 3. Plasma concentration of CKLP1 and levcromakalim following topical application of CKLP1 to the eye.**
CKLP1 was added to both eyes of Dutch-belted pigmented rabbits and plasma was isolated at various time points after the daily dose on days 1, 4 and 8. Both CKLP1 and its parent compound levcromakalim were detected in the plasma and data was used to calculate the standard pharmacokinetic parameters for these days, shown in Table 2. Data represented in log scale with a moving average trend line.

to day 1 (20.0 ± 6.3 for CKLP1 and 0.4 ± 0.1 ng/ml for levcromakalim), although these differences were not statistically significant (p = 0.47 for levcromakalim; p = 0.18 for CKLP1) (Table 2). $T_{max}$ values were found to be shortest on day 8, with 20 ± 8.7 minutes for CKLP1 and 140 ± 91.7 minutes for levcromakalim (Table 2). $T_{max}$ and AUC values did not show a statistically significant difference, although positive trends were observed between early and later days (Table 2). After adjusting for doses, comparison of AUC values between intravenous and topical treatment groups show bioavailability values for CKLP1 of 3.21, 4.26 and 4.03 for days 1, 4 and 8 of topical treatment. For levcromakalim, the bioavailability was determined to be 0.07, 0.13, and 0.12 for days 1, 4 and 8, respectively.

To examine whether CKLP1 and levcromakalim could be detected in the rabbit eye after treatment, we isolated aqueous humor and vitreous humor 24 h following the last blood draw on days 1, 4 and 8. No CKLP1 was detected in aqueous or vitreous humor, but low levels of levcromakalim were identified on all 3 days. Levcromakalim concentrations were reproducible and appeared to increase slightly following multiple days of treatment, similar to that observed

**Table 2. Pharmacokinetic parameters of CKLP1 and levcromakalim in rabbit plasma following administration of eye drops containing CKLP1.**

| | CKLP1 | | | | | | Levcromakalim | | | | | |
|---|---|---|---|---|---|---|---|---|---|---|---|---|
| | Half-life (min) | $T_{max}$ (min) | $C_{max}$ (ng/ml) | $T_{last}$ (h) | $C_{last}$ (ng/ml) | $AUC_{last}$ (ng/ml) | Half-life (min) | $T_{max}$ (min) | $C_{max}$ (ng/ml) | $T_{last}$ (h) | $C_{last}$ (ng/ml) | $AUC_{last}$ (ng/ml) |
| **Day 1** | 10.7 ± 4.1 | 360.0 ± 207.9 | 20.0 ± 6.3 | 24.0 ± 0.0 | 5.1 ± 1.1 | 271.3 ± 55.0 | 13.2 ± 2.6 | 400.0 ± 138.6 | 0.4 ± 0.1 | 24.0 ± 0.0 | 0.1 ± 0.1 | 6.1 ± 2.0 |
| **Day 4** | 6.7 ± 2.0 | 325.0 ± 268.5 | 30.6 ± 7.3 | 24.0 ± 0.0 | 4.2 ± 2.6 | 359.7 ± 82.0 | 25.8 ± 13.7 | 200.0 ± 69.3 | 0.9 ± 0.7 | 24.0 ± 0.0 | 0.3 ± 0.1 | 10.9 ± 6.0 |
| **Day 8** | 7.7 ± 3.0 | 20.0 ± 8.7 | 36.3 ± 13.3 | 24.0 ± 0.0 | 4.2 ± 1.5 | 340.9 ± 45.2 | 12.2 ± 5.6 | 140.0 ± 91.7 | 0.6 ± 0.3 | 18.7 ± 9.2 | 0.2 ± 0.1 | 10.2 ± 0.3 |

in plasma (Fig 4). However, no statistical significance was observed (levcromakalim in aqueous humor of right eye, p = 0.36; levcromakalim in aqueous humor of left eye, p = 0.05; levcromakalim in vitreous humor of right eye, p = 0.35; levcromakalim in vitreous humor of left eye, p = 0.32). Treatments appeared to be well-tolerated by all animals with no visible signs of toxicity or discomfort at any time during or after the treatment period.

## Tissue distribution of CKLP1 and levcromakalim following 90 days of daily topical ocular instillation

To assess the ocular tissue distribution of CKLP1 and levcromakalim, we treated Dutch-belted pigmented rabbits daily with a 50 μl eye drop of 10 mM CKLP1 for 90 consecutive days. Efficacy of CKLP1 treatment was assessed by IOP, which showed a 2.4 ± 0.4 mmHg absolute change in treated eyes compared to control eyes (p<0.001, n = 6, Fig 5). This corresponded to a 14.1 ± 1.0% reduction in the CKLP1 treated eye compared to the contralateral vehicle treated control eye, consistent with previous reports in short term CKLP1 treatment rabbit studies [12, 13]. In comparison to baseline values, IOP in the treated eye dropped by 11.4 ± 4.7% (baseline 16.5 ± 0.7 vs. treated 14.6 ± 1.0 mm Hg, p = 0.002). Animals were sacrificed either 24 hours after treatment (n = 3; treatment group) or after 72 hours (n = 3; washout group). Average IOP in treated eyes of the washout group (n = 3) was similar to pre-treatment baseline (baseline 16.5 ± 0.7 vs. post treatment 15.6 ± 0.6 mm Hg, p = 0.26, n = 3). Analysis of plasma showed no detectable levels of CKLP1 or levcromakalim 24 hours after the last treatment on Day 90. Interestingly, a consistent low level of levcromakalim was found in aqueous (0.2 ± 0.1 ng/ml; 3 animals of treatment group and 1 animal of washout group) and vitreous humor (0.1 ± 0.0 ng/ml, 3 animals of treatment group). CKLP1 was undetectable in all animals of both groups. Measurement of CKLP1 at 24 h following the last treatment in brain, heart,

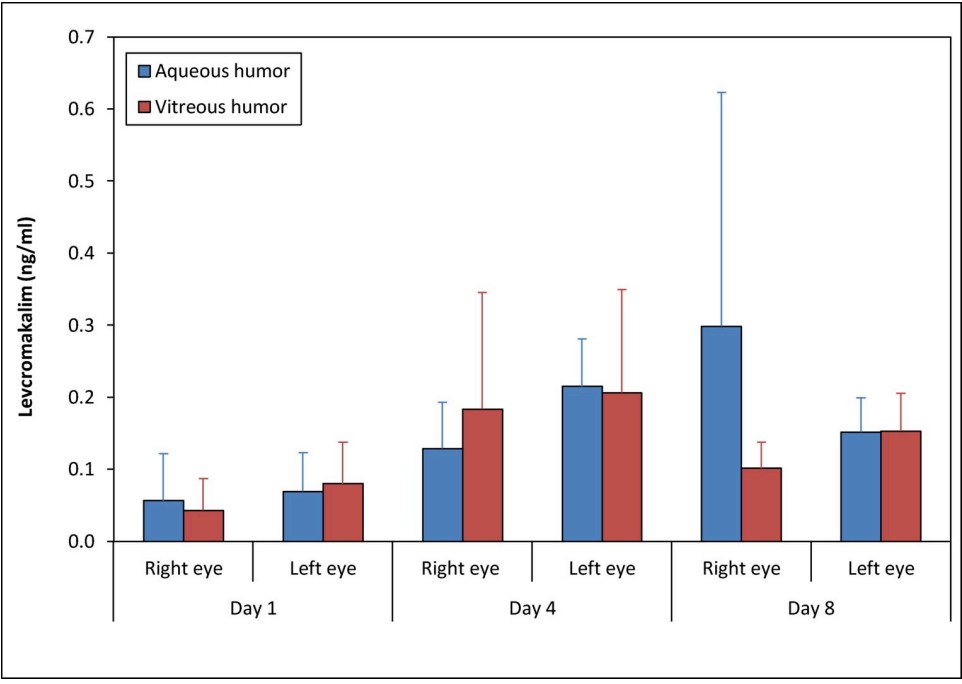

**Fig 4. Concentration of levcromakalim in aqueous and vitreous humor of rabbits following topical CKLP1 application.** Analysis of rabbit aqueous and vitreous humor following topical application of CKLP1 to both eyes showed presence of levcromakalim, but CKLP1 concentration was below detectable limits.

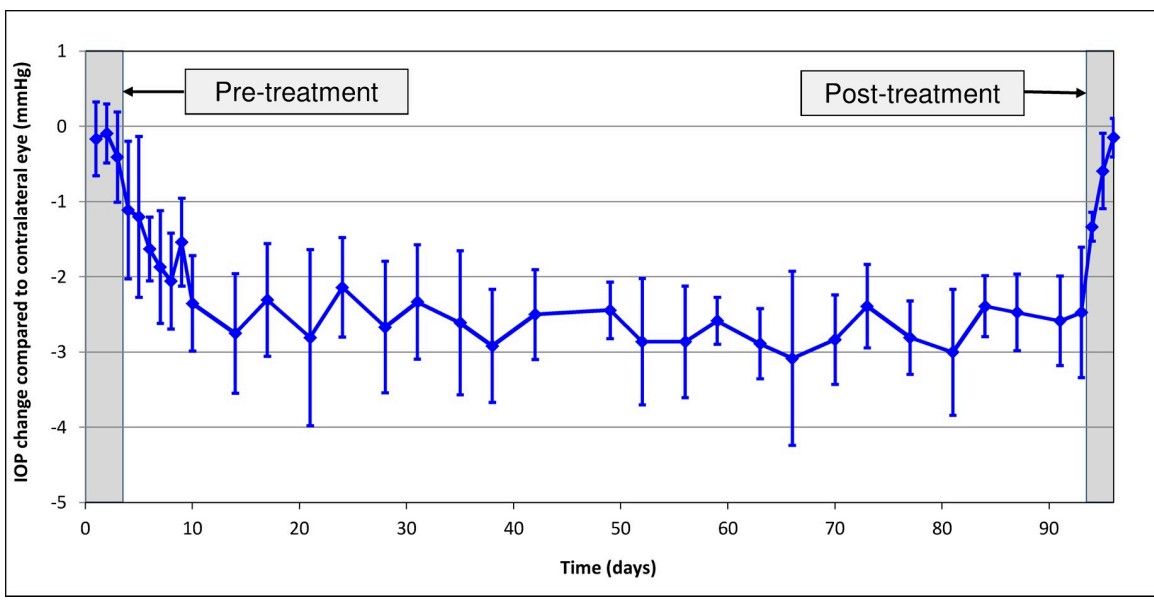

**Fig 5. Dutch-belted pigmented rabbits maintain IOP reduction following long term treatment with CKLP1.** CKLP1 was added to one eye while the contralateral eye received vehicle. IOP was measured to functionally validate the long term ocular topical treatment of CKLP1. On average, IOP was lowered by 14.1 ± 1.0% compared to vehicle treated eye, for the entire duration of treatment.

kidney, liver, lung and skeletal muscle showed detectable levels only in the kidney (15.8 ±1.8 ng/g, all animals of treatment and washout groups). Levcromakalim was found in lung (1.5 ± 0.0 ng/g, all animals of treatment and washout groups) and 4 out of the 6 rabbits showed levcromakalim in the liver (average 5.9 ± 4.3 ng/g, 3 animals of treatment group and 1 animal of washout group). CKLP1 and levcromakalim were not detected in the brain, heart and skeletal muscle of any animals in the treatment and washout groups (Table 3).

To assess cell and tissue morphology following CKLP1 treatment for 90 days, necropsy was performed on all 6 treated rabbits. Representative hematoxylin and eosin stained sections from 34 tissues were isolated, processed and assessed in a masked manner by a certified veterinary pathologist. Fig 6 shows representative images from trabecular meshwork, cornea, liver and kidney. No observable histological changes were noted between treated and control tissues. Additional images from all examined tissues can be found in supplementary figure (S1 Fig).

## Discussion

In the current study, once daily administered CKLP1 was successfully converted to levcromakalim and can be detected in aqueous humor, vitreous humor, plasma and select tissues

**Table 3. Concentration of CKLP1 and levcromakalim in selected rabbit tissues and fluids following CKLP1 treatment with topical eye drops for 90 consecutive days.**

|  | Plasma (ng/ml) | Brain (ng/g) | Heart (ng/g) | Kidney (ng/g) | Liver (ng/g) | Lung (ng/g) | Muscle (ng/g) | Aqueous humor (ng/ml) | Vitreous humor (ng/ml) |
|---|---|---|---|---|---|---|---|---|---|
| **CKLP1** | ND | ND | ND | 15.8 ± 1.8 (n = 6/6) | ND | ND | ND | ND | ND |
| **Levcromakalim** | ND | ND | ND | ND | 5.9 ± 4.3 (n = 4/6) | 1.5 ± 0.0 (n = 6/6) | ND | 0.2 ± 0.1 (n = 4/6) | 0.1 ± 0.0 (n = 3/6) |

ND–Not detectable.

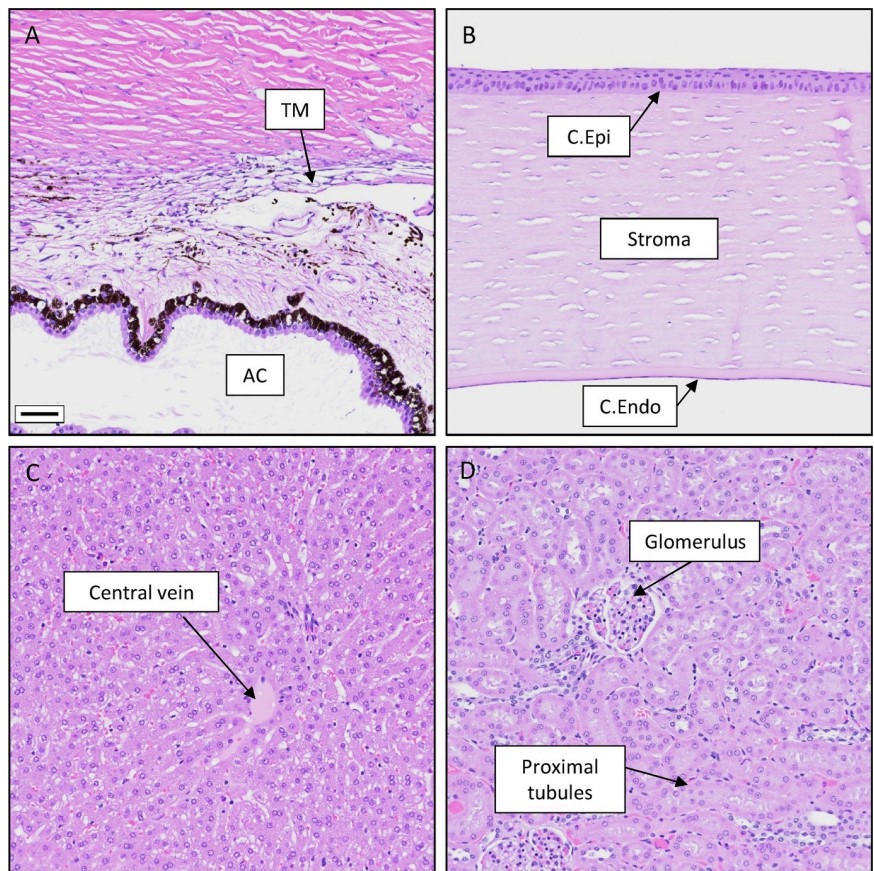

**Fig 6. Representative rabbit tissue sections following long term CKLP1 treatment.** Following 90 days of CKLP1 treatment, rabbit tissues did not show any observable histological changes. A. Trabecular meshwork (TM) and the anterior chamber (AC) B. Cornea showing the epithelium (C. Epi) and endothelium (C. Endo). C. Liver, D. Kidney. Scale bar, 50 μm. Additional tissue sections can be found in S1 Fig.

following intravenous or topical application to the eye. Both CKLP1 and levcromakalim are eliminated from the plasma with relatively short terminal half-lives.

Dutch-belted pigmented rabbits have been used to perform pharmacokinetic studies for several FDA-approved glaucoma medications [15, 16]. The Dutch-belted rabbit is a suitable species for evaluating ocular and systemic tissue distribution, and this model can also provide quantitative ocular and plasma pharmacokinetic data. Additionally, the pigmented rabbit enables the examination of compound's interaction with melanin which more closely represents an environment similar to humans. We also have previously reported that short term (5 days) topical eye treatment with CKLP1 effectively lowered IOP in Dutch-belted pigmented rabbits [13]. For these reasons, Dutch-belted pigmented rabbits seemed like a suitable choice to perform the first pharmacodynamics studies for CKLP1.

Although we envision CKLP1 as an ocular topical drug for IOP reduction, we initially validated our method for detecting CKLP1 and levcromakalim in plasma following intravenous injections. Since intravenous administration would create a much higher systemic exposure of compound and its active metabolite in plasma (compared to topical administration), this group served as a reference, providing necessary information with regard to CKLP1 to levcromakalim conversion, clearance and systemic availability. The data obtained from our study suggests that the levcromakalim found in plasma was a result of the conversion of CKLP1.

Plasma levels of levcromakalim were found to increase over the first hour, in contrast to CKLP1.There does not appear to be a complete conversion of CKLP1 to levcromakalim. In comparison to $C_{max}$ values, levcromakalim peak concentrations are almost 200 fold less than CKLP1. While this may be under-estimated due to our limited time point collections, it is also possible that the converted product is either rapidly internalized into tissue or efficiently eliminated from the body making detection in plasma difficult. Because the current study was designed to evaluate the pharmacokinetic parameters of the compounds and not the exact metabolic/absorption site of the drugs, future studies will be necessary to determine the metabolic profile along with tissue distribution of the drug.

The ultimate goal is to move CKLP1 into the clinic where it can be used to benefit patients with ocular hypertensive disorders such as glaucoma. We did not detect quantifiable levels of CKLP1 or levcromakalim in the aqueous humor or vitreous of rabbits treated with CKLP1 daily for 8 consecutive days. This may be a consequence of not isolating aqueous humor until 24 h after last dose. Since aqueous humor has approximately a 2 h half-life, it is conceivable that most CKLP1 would have been removed from the anterior segment. On the other hand, levcromakalim was found in both aqueous and vitreous humor at detectable concentrations. This suggests that phosphatases present in the cornea and anterior chamber of the eye appear to have cleaved the phosphate group from CKLP1 and converted it into levcromakalim.

In our short term topical group (CKLP1 added once daily for 8 days), we found detectable levels of both CKLP1 and levcromakalim in plasma. This was expected as drugs applied topically to the eye will get into the aqueous humor and be removed from the anterior chamber by the normal outflow pathways. The outflow pathways connect to episcleral veins and eventually the aqueous humor and its contents are absorbed into the systemic venous circulation. Additionally, because the rabbit has an extensive retro bulbar plexus, a portion of the runoff from the topical dose has the potential to directly enter into the blood stream. However, both CKLP1 and levcromakalim concentrations in plasma were significantly lower than what we found in the intravenous administration group. Like the intravenous treated group, maximal levcromakalim concentrations lagged behind maximal CKLP1 concentrations, suggesting conversion of CKLP1 to levcromakalim. However due to small sample size, additional experiments will need to be performed to specifically address elimination efficiency and examine whether CKLP1 or levcromakalim may accumulate in ocular tissues. These studies in a large animal model are currently underway in our laboratory.

It should also be noted that various pharmacokinetic parameters such as higher peak CKLP1 and levcromakalim concentrations at days 4 and 8 compared to day 1, as well as the amount of levcromakalim detected in the ocular fluids, were not statistically significant. This is not surprising since the current study was not designed to determine significance across days but to evaluate if there were any trends. These results provide us with the level of differences in drug concentrations we can expect following repeated treatments. Experiments designed to test accumulation of drugs in ocular fluids and tissues will require a larger study that is correctly powered.

Topical ocular instillation of Dutch-belted pigmented rabbits for 90 days with CKLP1 confirmed results from our short term study where CKLP1 was not detected in aqueous or vitreous humor at 24 h after administration. However, levcromakalim was identified in aqueous humor and vitreous humor in all rabbits that were sacrificed 24 hours after last treatment (3 out 3). Only 1 animal of the washout group showed levcromakalim in aqueous humor. Interestingly, CKLP1 and levcromakalim were not detectable in the plasma of any of the 6 animals treated with CKLP1 for 90 days. While the exact reason for the absence of these drugs in plasma is unclear, the absence of active compounds after 90 days of treatment suggests an efficient elimination mechanism of the prodrug and its parent compound. In concert with this,

CKLP1 appears to be well tolerated by the rabbits as no observable histological changes were noted in any of the studied tissues after 90 days of once-daily treatment.

In summary, CKLP1 appears to be a viable prodrug of the $K_{ATP}$ channel opener levcromakalim. When applied topically to the eye, CKLP1 is converted into its active parent compound levcromakalim which results in significant IOP reduction in normotensive Dutch-belted pigmented rabbits.

## Supporting information

**S1 Fig. Representative images of rabbit tissues isolated during necropsy following once daily CKLP1 treatment with eye drops for 90 days.** No observable differences in histology were noted between vehicle-treated and CKLP1 treated tissues as evaluated by a masked veterinary pathologist following 90 days of treatment with CKLP1 eye drops. A, adrenal gland; B, Aorta; C, bone marrow; D, brain; E, cecum; F, colon; G, duodenum; H, esophagus; I, gall bladder; J, heart; K, ileum; L, jejunum; M, lung; N, mammary gland; O, muscle; P, ovary; Q, pancreas; R, pituitary gland; S, salivary gland; T, sciatic nerve; U, skin; V, spinal cord; W, spleen; X, stomach; Y, thymus; Z, thyroid; AA, trachea; BB, urinary bladder; CC, uterus; DD, vagina. Scale bar, 50 μm.
(TIF)

## Author Contributions

**Conceptualization:** Uttio Roy Chowdhury, Michael P. Fautsch.

**Data curation:** Uttio Roy Chowdhury, Rachel A. Kudgus, Tommy A. Rinkoski, Bradley H. Holman, Cindy K. Bahler, Cheryl R. Hann, Peter I. Dosa.

**Formal analysis:** Uttio Roy Chowdhury, Rachel A. Kudgus, Joel M. Reid, Michael P. Fautsch.

**Funding acquisition:** Peter I. Dosa, Michael P. Fautsch.

**Investigation:** Uttio Roy Chowdhury.

**Methodology:** Uttio Roy Chowdhury, Rachel A. Kudgus, Tommy A. Rinkoski, Bradley H. Holman, Cindy K. Bahler, Cheryl R. Hann, Joel M. Reid, Peter I. Dosa, Michael P. Fautsch.

**Project administration:** Uttio Roy Chowdhury, Michael P. Fautsch.

**Resources:** Michael P. Fautsch.

**Supervision:** Joel M. Reid.

**Writing – original draft:** Uttio Roy Chowdhury, Rachel A. Kudgus, Michael P. Fautsch.

**Writing – review & editing:** Uttio Roy Chowdhury, Rachel A. Kudgus, Tommy A. Rinkoski, Bradley H. Holman, Cindy K. Bahler, Cheryl R. Hann, Joel M. Reid, Peter I. Dosa, Michael P. Fautsch.

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
