## [Decision Letter · Decision Letter 0]

5 Mar 2020

PONE-D-20-00899

Pharmacological and pharmacokinetic profile of the novel ocular hypotensive prodrug CKLP1 in Dutch-belted pigmented rabbits

PLOS ONE

Dear Dr Fautsch,

Thank you for submitting your manuscript to PLOS ONE. After careful consideration, we feel that it has merit but does not fully meet PLOS ONE’s publication criteria as it currently stands. Therefore, we invite you to submit a revised version of the manuscript that addresses the points raised during the review process.

Several concerns, clearly detailed in reviews, should be carefully addressed.

We would appreciate receiving your revised manuscript by Apr 19 2020 11:59PM. To enhance the reproducibility of your results, we recommend that if applicable you deposit your laboratory protocols in protocols.io, where a protocol can be assigned its own identifier (DOI) such that it can be cited independently in the future. For instructions see: http://journals.plos.org/plosone/s/submission-guidelines#loc-laboratory-protocols

We look forward to receiving your revised manuscript.

Kind regards,

Ted S Acott, PhD

Academic Editor

PLOS ONE

Journal Requirements:

2. To comply with PLOS ONE submissions requirements, please provide methods of sacrifice in the Methods section of your manuscript.

Reviewers' comments:

Reviewer's Responses to Questions

**Comments to the Author**

1. Is the manuscript technically sound, and do the data support the conclusions?

Reviewer #1: Yes

Reviewer #2: Yes

2. Has the statistical analysis been performed appropriately and rigorously? 

Reviewer #1: No

Reviewer #2: Yes

3. Have the authors made all data underlying the findings in their manuscript fully available?

Reviewer #1: Yes

Reviewer #2: Yes

4. Is the manuscript presented in an intelligible fashion and written in standard English?

Reviewer #1: Yes

Reviewer #2: Yes

5. Review Comments to the Author

Reviewer #1: Chowdhury et al present the pharmacological and pharmacokinetic data of a novel prodrug, CKLP1. Overall the study was well designed and demonstrates the necessary profile of this novel drug as a potential therapeutic target. The following suggestions would improve the manuscript:

1. In Figure 2 the authors reference the half-life CKLP1 and Levcromakalim. Can this be highlighted better in the figure? The number is referenced in the text but it would be useful to denote the half-life on the graph as well.

2. Lines 191-192 references Figure 3 and the peak concentration comparisons between days. Can this data be statistically compared?

3. Figure 4, Can this data also be statistically compared (Line 214?

4. Provide more detailed methods on how IOP was measured and what statistical analysis was performed on these data.

Reviewer #2: PONE-D-20-0899

Summary

In this well-written article, the authors study chromakalim prodrug 1(CKLP1) and its derivative, levcromakalim. These drugs can reduce IOP which is important because there is elevated intraocular pressure (IOP) in glaucoma. They focus on the pharamcokinetics of these drugs in Dutch-Belted pigmented rabbits. In this paper, the authors can detect both molecules using LC-MS/MS. In an CKLP1 intravenous delivery model, they measure pharmacokinetic parameters in the blood. The salient findings are that both molecules are detected with levcromakalim coming on slower, suggesting that CKLP1 is converted to it. Then, there are two topical CKLP1 delivery paradigms. The first is short term with both eyes of the rabbits receiving topical drug for 1-, 4-, or 8-days. The second is a longer experiment with one eye receiving topical drug (the other eye served as a vehicle control) for 90-days. Blood, organs, and intraocular contents (aqueous and vitreous) were collected for analyses at various time points. In the,

1-, 4-, and 8-day experiments CKLP1 was not detected in the aqueous or vitreous.

1-, 4-, and 8-day experiments levcromakalim was detected in the aqueous and vitreous.

In the 90-day experiment CKLP1 was not detected in the aqueous or vitreous

In the 90-day experiment levcromakalim was sometimes detected in the aqueous and vitreous.

In the 90-day experiment neither were detected in plasma

Further, the 90-day topical drug study did not lead to observable histological changes to the eye and multiple other organs.

Ultimately, there are some question about the methods and about where the drug goes after topical delivery.

Comments:

1) The authors note that IOP is reduced by 2.4 +/- 0.4 mm Hg and 14.1 +/- 1% compared to the other eye of the same rabbit. While this is nice, what was the IOP compared to the baseline IOP in the eye that was actually treated? In other words, what was the magnitude and percent of IOP reduction in the eye that was treated, pre- and post-? Getting data from both eyes (pre- and post- drug or vehicle) controls for more confounders.

2)While the Cmax for CKLP1 and levcromakalim may not have been significantly different in the 1- to 8-day experiments (Table 2), it appears that the Tmax might be? Do that calculation and include/implement. It is interesting to me because it says that the loading is faster. Makes sense given that the drop has been there for 7 days prior. The AUCs too.

3)If chromakalim (which is broken down to a dex- and lev-type) can lower IOP (citation 13), what is the advantage or need for CKLP1? Because it is already the lev-type? If one understood the breakdown of chromakalim and ratio of dex- and lev- that is produced, all one has to do is to upscale the dose of chromakalim to get more lev-. Further, the dex- had some IOP lowering effect too (citation 13). Unless, the dex- has some potential side-effect?

4)For the topical experiments, it is slightly strange that absolutely no CKLP1 is found in the aqueous and vit.

The authors suggest that testing the levels 24 hours after the last drop may have caused the aqueous turnover to make the drug leave the eyes (lines 285-298). I have a few thoughts.

-First, topical drugs have a much more direct route to the systemic body than aqueous outflow. Topical drugs simply enter the nasolacrimal duct system (like how we taste our salty tears when we cry). The drug can then be picked up by the blood vessels in the oropharynx and nasal mucosal epithelium. This alone can explain presence of drugs in the blood stream after topical delivery in the short-term experiments. This is also an issue for comment #5 below.

-Second, maybe the drug does not even have to get into the eye. The authors maybe testing the aqueous and vitreous under the assumption that that drugs must move into the eye to impact IOP-governing mechanisms. However, as the authors note, these drugs are supposed to effect the distal outflow pathways (lines 59-60). Thus, could the drug just penetrate into the scleral tissue and directly impact outflow as opposed to entering the anterior chamber first to do so?

-Third, the methods involve taking the aqueous/vit/plasma, spinning it down and testing the free fractions. What if CKLP1 is pigment bound? Beta blockers can be. In this case, the iris may have absorbed it all. Can we test for drug in iris? Try homogenization iris followed by a fast spin and testing what is in the sup.

-Fourth, what if there is differential protein binding such that the drug in the vitreous cavity was actually stuck with the vitreous that had not undergone syneresis? To address this, the pellet could be collagenase treated and spun again with the sup tested.

-Lastly, maybe the phosphatases are so fast that all the CKLP1 was metabolized.

5)Then for the topical experiments, it is strange that after 90 days, no levels were detected in the plasma. This is quite strange because both drugs were detected systemically after the shorter experiments at the 24-hour time-point. Further in 2 of 6 animals, no levcromakalim was seen in the aqueous while in 3 of 6 animals no levcromakalim was seen in the vitreous. This was unlike the shorter topical experiments were levcromakalim was more consistently detectable. Thus, the drug has to be somewhere. The renal/liver data suggests some elimination mechanism but this seems not enough. Since the drops applied on day 88 and 89 aren’t enough to show drug in the plasma, either the clearance method is revved up or there a depot where the drug is just accumulating. My feeling is that the drug is going somewhere that we are not recognizing. As above, I would look for the drug in other ocular tissues like the actual vitreous itself (you are only testing the liquified portion) and the iris. Also, in the blood fraction, test for binding. One could repeat the 90-day experiment, collect the blood and test for the drug in the packed RBC portion (after the spin). Alternatively, just test for in vitro binding by getting rabbit whole blood, mixing it with drug, waiting for a period of time, spinning the blood down and testing for the drug in the plasma vs. packed RPC portion. At least, this in vitro experiment could show if the drug can get into or onto RBCs and then help solve some of the mystery.

Minor issues:

1)lines 139… for this section, you must mean “for rabbits treated with CKLP1 for up to 8 days..” as opposed to just 8 days since some rabbits received treatment for 1- and 4-days.

2)lines 301: levcromakalim cannot be “consistently” identified if it is only seen in 3 of 6 rabbits for the vitreous and in 4 of 6 rabbits in the aqueous.

6. PLOS authors have the option to publish the peer review history of their article (what does this mean?). If published, this will include your full peer review and any attached files.

Reviewer #1: No

Reviewer #2: No

---

## [Decision Letter · Decision Letter 1]

2 Apr 2020

Pharmacological and pharmacokinetic profile of the novel ocular hypotensive prodrug CKLP1 in Dutch-belted pigmented rabbits

PONE-D-20-00899R1

Dear Dr. Fautsch,

We are pleased to inform you that your manuscript has been judged scientifically suitable for publication and will be formally accepted for publication once it complies with all outstanding technical requirements.

With kind regards,

Ted S Acott, PhD

Academic Editor

PLOS ONE

Additional Editor Comments (optional):

concerns addressed

Reviewers' comments:

Reviewer's Responses to Questions

**Comments to the Author**

1. If the authors have adequately addressed your comments raised in a previous round of review and you feel that this manuscript is now acceptable for publication, you may indicate that here to bypass the “Comments to the Author” section, enter your conflict of interest statement in the “Confidential to Editor” section, and submit your "Accept" recommendation.

Reviewer #1: All comments have been addressed

Reviewer #2: All comments have been addressed

2. Is the manuscript technically sound, and do the data support the conclusions?

Reviewer #1: (No Response)

Reviewer #2: Yes

3. Has the statistical analysis been performed appropriately and rigorously? 

Reviewer #1: (No Response)

Reviewer #2: Yes

4. Have the authors made all data underlying the findings in their manuscript fully available?

Reviewer #1: (No Response)

Reviewer #2: Yes

5. Is the manuscript presented in an intelligible fashion and written in standard English?

Reviewer #1: (No Response)

Reviewer #2: Yes

6. Review Comments to the Author

Reviewer #1: (No Response)

Reviewer #2: The authors have attempted additional statistical analyses that I requested.

The authors have explained my misunderstanding of the drug

The authors have expanded their discussion as to alternative ways the drug could enter the body and given a better explanation as to why levcromakalim was only seen in 3 of 6 animals

7. PLOS authors have the option to publish the peer review history of their article (what does this mean?). If published, this will include your full peer review and any attached files.

Reviewer #1: No

Reviewer #2: No

---

## [Editor Report · Acceptance letter]

6 Apr 2020

PONE-D-20-00899R1 

Pharmacological and pharmacokinetic profile of the novel ocular hypotensive prodrug CKLP1 in Dutch-belted pigmented rabbits 

Dear Dr. Fautsch:

I am pleased to inform you that your manuscript has been deemed suitable for publication in PLOS ONE. Congratulations! Your manuscript is now with our production department. 

With kind regards,

on behalf of

Dr. Ted S Acott 

Academic Editor

PLOS ONE